# Effective Influence Maximization with Priority

## Abstract

Influence maximization (IM) aims to identify a small set of influential users to maximize the information spread. It has been widely applied in the context of viral marketing, where a company distributes incentives to a few influencers to promote the product. However, in practical scenarios, not all users hold equal importance and certain users need to be prioritized for the specific requirements. Motivated by this, recently, a variant problem of IM, called *influence maximization with priority* (IMP), has been proposed. Given a graph $G = (V, E)$, a priority set $P \subseteq V$ and a threshold $T \in [0, |P|]$, IMP aims to identify a set of $k$ nodes (termed *seeds*) to maximize the expected number of activated nodes in $G$ while satisfying that the expected number of activated nodes in $P$ is no less than the given threshold. Nevertheless, we show that existing solutions for IMP are inferior in maximizing the influence spread in $G$, and can only offer poor approximation ratios in many cases. To address these limitations, in this paper, we first propose a novel framework named SAR with both superior empirical effectiveness and strong theoretical guarantees. In addition, to obtain more practical results, we study the IMP problem under the *adaptive* setting, where the seed users are iteratively selected after observing the diffusion result of the previous seeds. We design a scalable and effective algorithm AAS that achieves expected approximation guarantees. Comprehensive experiments on 5 real-world datasets are conducted to validate the performance of the proposed techniques. Compared with the state-of-the-art method, SAR achieves up to 22.3% larger spread and AAS achieves up to 42.6% larger spread, with both exhibiting a higher empirical approximation ratio.

## CCS Concepts

• **Do Not Use This Code** → **Generate the Correct Terms for Your Paper**; *Generate the Correct Terms for Your Paper*; Generate the Correct Terms for Your Paper; Generate the Correct Terms for Your Paper.

**ACM Reference Format:**
Anonymous Author(s). 2018. Effective Influence Maximization with Priority. In *Proceedings of Make sure to enter the correct conference title from your rights confirmation emai (Conference acronym 'XX).* ACM, New York, NY, USA, 11 pages. https://doi.org/XXXXXXX.XXXXXXX

## 1 Introduction

Given a graph $G = (V, E)$ and a positive integer $k$, the influence maximization (IM) problem aims to identify a set of $k$ nodes in $G$

that can influence as many users as possible [2, 4–6, 18, 28–30, 34]. This problem finds important applications in various fields, such as viral marketing [9, 25], network monitoring [20] and rumor controlling [3, 32]. Among these, viral marketing is the most representative scenario from which IM originates. In this context, the company promotes a product by distributing incentives (e.g., free samples) to a set of influential users, in hopes of creating a large cascade of product adoptions via the word-of-mouth effect.

However, IM neglects the priority of individuals, hindering its applicability in real-world marketing scenarios. Specifically, not all users are of equal importance, and certain users need to be prioritized for specific requirements. For example, when promoting gaming equipment, the company typically prioritizes potential customers (e.g., avid gamers or technology enthusiasts) who frequently engage in gaming activities and are well-suited for the product. Motivated by this, Pham et al. [24] formulate the *influence maximization with priority* (IMP) problem. Given a graph $G = (V, E)$, a priority set $P \subseteq V$, a threshold $T \in [0, |P|]$ and a budget $k \geq T$, the IMP problem aims to identify a seed set $S$ of $k$ nodes that maximizes $\mathbb{E}[I(S)]$ while satisfying $\mathbb{E}[I_P(S)] \geq T$ (termed as *threshold condition*), where $\mathbb{E}[I(S)]$ (resp. $\mathbb{E}[I_P(S)]$) is the expected number of nodes activated by $S$ in $G$ (resp. $P$). Note that, the assumption $k \geq T$ is to ensure the threshold condition can always be satisfied.

The IMP problem is NP-hard and it cannot be approximated within a ratio of $1 - 1/e + \epsilon$ for any $\epsilon > 0$ unless P = NP. The state-of-the-art approach for IMP provides a data-dependent approximation and consists of two stages generally [24]. In the first stage, it iteratively selects the node that maximizes the influence spread in $P$ until the threshold condition is satisfied. In the second stage, in a similar manner to the state-of-the-art solution for IM [28], it iteratively selects the seed node that maximizes the influence spread in $G$ until the budget is exhausted. Nevertheless, we show that IGS is suboptimal in maximizing the influence spread in $G$ and can only yield a poor approximation ratio theoretically. The reason is that, due to the possibility that the seeds selected in the second phase could influence the prioritized nodes, it may not be necessary for the first stage to fully satisfy the threshold condition. Therefore, we can allocate a smaller budget to the first stage, reserving more for the second stage, which can result in a larger influence spread in $G$. However, in the practical implementation, determining the ideal budget for the first stage in advance is challenging.

To tackle these limitations, in this paper, we propose a novel framework called U̲nderline Select-A̲nd-R̲eplace (SAR) with superior empirical effectiveness for the IMP problem. In general, SAR consists of two stages, i.e., select stage and replace stage. In the select stage, SAR identifies a size-$k$ seed set $S$ that maximizes the influence spread in $G$ by applying the greedy strategy. Note that, to provide tight theoretical guarantees for the final result, different from the conventional IM problem, we need to ensure that the set returned in each iteration (within the greedy strategy) can all provide theoretical guarantees. In the replace stage, SAR processes the seeds in $S$ according to the reverse order of their insertion order in the

select stage. For each seed in $S$, SAR first removes it from $S$, and then selects the node that maximizes the influence spread in $P$ and adds it to $S$. This replacement process stops until the threshold condition is satisfied. Theoretically, SAR can offer a provable $(1 - (1 - 1/k)^{k_1^b} - \epsilon)$-approximate solution, where $k_1^b$ is the actual budget within the solution of SAR that maximizes the influence spread in $G$, i.e., $k_1^b$ equals $k$ minus the number of nodes replaced in the second phase. Compared to IGS, SAR achieves larger influence spread and offers a higher empirical approximation ratio.

As introduced above, IGS and SAR both focus on the *non-adaptive* setting, which requires all seeds to be selected in one batch without making any observation on the actual influence spread. However, such a setting fails to take advantage of the previous spreading results when selecting the next seed node, which may result in the activation of the same node multiple times, ultimately leading to the inferior influence spread. The *adaptive* strategy, where the seeds are iteratively selected after observing the diffusion result of the previous seeds, has been shown to be more effective in real-world cases [1, 7, 8]. Motivated by this, to obtain more practical results, we further study the IMP problem under the adaptive setting. For this purpose, an intuitive idea is to directly extend SAR to the adaptive setting. However, it is infeasible for us to implement the replacement procedure in an adaptive manner, since under the adaptive setting, before we select the next seed, the propagation of the previously selected seeds has been finalized. Therefore, we cannot *regret* the selection of a node and then choose another one. To address this issue, we design a novel framework named Adaptive-Alternation-Selection (AAS) with a $(1 - e^{(\epsilon-1) \cdot k^c/k})$-expected approximation, where $k^c$ is the actual budget within the solution of AAS that maximizes the influence spread in $G$. Specifically, for each node selection, based on a judgment condition, AAS adopts one of two proposed procedures, each aiming to identify a seed that maximizes the influence spread in either $G$ or $P$. Afterwards, it observes the newly activated nodes and updates the corresponding information, to prevent these nodes from being repeatedly activated in subsequent processes. Note that, due to the novel alternating selection mechanism, the theoretical results of existing adaptive solutions (e.g., adaptive IM [12–14]) cannot be applied to AAS, rendering the derivation of its approximation guarantee particularly challenging. Finally, experiments over 5 real-world datasets are conducted to verify the performance of proposed algorithms. The main contributions of the paper are summarized as follows.

- We propose a novel framework SAR that returns a $(1 - (1 - 1/k)^{k_1^b} - \epsilon)$-approximate solution for the IMP problem, where $k_1^b$ is the actual budget within the solution of SAR that maximizes the influence spread in $G$. (Section 3)
- We conduct the first research to study the IMP problem under the adaptive setting, and design an effective framework AAS that returns a $(1 - e^{(\epsilon-1) \cdot k^c/k})$-expected approximate solution, where $k^c$ is the actual budget within the solution of AAS that maximizes the influence spread in $G$. (Section 4)
- We conduct extensive experiments on 5 real-world graphs to verify the performance of proposed techniques. Compared with the state-of-the-art method, SAR and AAS both demonstrate better performance in terms of influence spread, and offer a higher empirical approximation ratio. (Section 5)

Note that, due to the limited space, all proofs are omitted and can be found in the Appendix A.

## 2 Preliminaries

In this section, we first formally define the *influence maximization with priority* (IMP) problem and analyze its hardness. Then, we present an overview of the existing solutions for IMP.

### 2.1 Problem Definition

We consider a social network as a directed graph $G = (V, E)$ with $|V| = n$ and $|E| = m$, where $V$ and $E$ represent the set of nodes and edges, respectively. Given an edge $\langle u, v \rangle \in E$, we refer to $u$ as an incoming neighbor of $v$ and $v$ as an outgoing neighbor of $u$. Each edge $\langle u, v \rangle$ is associated with a propagation probability $p(u, v) \in [0, 1]$, representing the probability that $u$ influences $v$.

**Diffusion model**. In this paper, we adopt the widely used *independent cascade* (IC) model [18] to simulate the propagation. Note that, the proposed techniques can be easily extended to support the *linear threshold* model. Given a seed set $S \subseteq V$, the diffusion process of $S$ under the IC model progresses in discrete timestamps, with the specifics described below.

- At timestamp 0, the nodes in the seed set $S$ are activated, while all other nodes remain inactive. Once a node is activated, it stays active in all subsequent timestamps.
- If a node $u$ becomes active at timestamp $t$, for each of its inactive outgoing neighbors $v$, $u$ has a single opportunity to activate $v$ with probability $p(u, v)$ at timestamp $t + 1$.
- The propagation process ends when no further nodes can be activated in the graph $G$.

Let $I_G(S)$ be the number of active nodes in $G$ at the end of the propagation process. On this basis, we use $\mathbb{E}[I_G(S)]$ to denote the *influence spread* of $S$ in $G$, where the expectation is taken over the randomness of propagation. For presentation simplicity, $G$ will be dropped when the context is clear. In [18], Kempe et al. propose the *live edge* procedure to characterize the diffusion process. Specifically, by removing each edge $\langle u, v \rangle \in E$ with $1 - p(u, v)$ probability, the remaining graph is referred to as a *realization* (denoted as $\phi$), based on which, the influence spread $\mathbb{E}[I(S)]$ of $S$ can be calculated below.

$$\mathbb{E}[I(S)] = \mathbb{E}_\Phi[I_\Phi(S)] = \sum_{\phi \in \Omega} I_\phi(S) \cdot p(\phi),$$

where $\Omega$ is the set of all possible realizations of $G$, $\Phi$ is a random realization sampled from $\Omega$, $p(\phi)$ is the probability for realization $\phi$ to occur, and $I_\phi(S)$ is the number of nodes reachable from $S$ in $\phi$.

**Problem statement**. Given a graph $G = (V, E)$, a priority set $P \subseteq V$, a threshold $T \in [0, |P|]$ and a budget $k \geq T$, the IMP problem aims to identify a size-$k$ seed set $S^*$ with the maximum influence spread $\mathbb{E}[I(S^*)]$ while satisfying $\mathbb{E}[I_P(S^*)] \geq T$, i.e.,

$$S^* = \underset{S \subseteq V, |S| \leq k}{\arg\max} \ \mathbb{E}[I(S)] \ \text{s.t.} \ \mathbb{E}[I_P(S)] \geq T,$$

where $\mathbb{E}[I_P(S)]$ is the expected number of nodes in $P$ activated by $S$ and we can say that $\mathbb{E}[I_P(S)]$ is the influence spread of $S$ in $P$.

It is clear that IMP will degenerate to the IM problem when $P = \emptyset$. As a consequence, the following lemmas hold trivially, illustrating the inherent complexity of the problem.

LEMMA 2.1. *The IMP problem is NP-hard and it cannot be approximated within a ratio of $1 - 1/e + \epsilon$ for any $\epsilon > 0$ unless $P = NP$.*

## 2.2 Existing Solutions Revisited

Here we revisit the state-of-the-art approaches for addressing the IMP problem and analyze their limitations. In the following, we first present the basic technique used for influence spread estimation.

**RIS method**. Given a seed set $S \subseteq V$, its influence spread cannot be computed within polynomial time [4]. To overcome this hurdle, Borgs et al. [2] propose the advanced *Reverse Influence Sampling* (RIS) method, which is based on the concept of *Reverse Reachable* (RR) set (denoted by $R$), and its generation is followed by two steps:

  *i)* sample a node $v$ uniformly at random from $V$.
  *ii)* perform a stochastic BFS from $v$ in the reverse directions of edges and store the visited node into $R$.

The randomly sampled node $v$ is referred to as the *source node* of $R$. Then, an unbiased estimator for $\mathbb{E}[I(S)]$ is derived with a set $\mathcal{R}$ of sufficient RR sets [2], i.e.,

$$\mathbb{E}[I(S)] = \mathbb{E}[n \cdot \frac{Cov_{\mathcal{R}}(S)}{|\mathcal{R}|}], \quad (1)$$

where $Cov_{\mathcal{R}}(S) = \sum_{R \in \mathcal{R}} \min\{|S \cap R|, 1\}$. The rationale behind this estimator is that the intersection between $S$ and an RR set indicates the potential influence of $S$ on the source node of the RR set. On the basis of RIS, we further introduce the concept of *Priority Reachable Reverse* (PRR) set. The only difference between the PRR set and RR set is that the source node of PRR set is uniformly at random selected from $P$. Given a priority set $P$ and a set $\mathcal{R}^P$ of PRR sets, an unbiased estimator for $\mathbb{E}[I_P(S)]$ can be derived similarly.

$$\mathbb{E}[I_P(S)] = \mathbb{E}[|P| \cdot \frac{Cov_{\mathcal{R}^P}(S)}{|\mathcal{R}^P|}]. \quad (2)$$

**The state-of-the-art approaches**. Pham et al. [24] propose two approximation algorithms for the IMP problem, named IG and IGS. In particular, IG can only be applied under the *value oracle* model, where the value of $\mathbb{E}[I(S)]$ and $\mathbb{E}[I_P(S)]$ is pre-given. This method is practically infeasible since these two functions are both #P-hard to compute. IGS is the scalable version for IG, which shares the same framework with IG, and the difference is that IGS employs the RIS method to approximate $\mathbb{E}[I(S)]$ and $\mathbb{E}[I_P(S)]$. Therefore, in the following, we only focus on IGS for presentation simplicity.

Let $\hat{I}(\cdot)$ (resp. $\hat{I}_P(\cdot)$) be the estimated value of $\mathbb{E}[I(S)]$ (resp. $\mathbb{E}[I_P(S)]$) via $\mathcal{R}$ (resp. $\mathcal{R}^P$), and $\hat{I}(u|S) = \hat{I}(u \cup S) - \hat{I}(S)$ (resp. $\hat{I}_P(u|S) = \hat{I}_P(u \cup S) - \hat{I}_P(S)$) be the marginal gain of adding $u$ to the set $S$ w.r.t. $\hat{I}(\cdot)$ (resp. $\hat{I}_P(\cdot)$). Generally, IGS consists of two stages and starts with an empty set $S$. In the first stage, it iteratively selects the seed $u$ that leads to the largest $\hat{I}_P(u|S)$ and adds it to $S$, until the threshold condition is satisfied, i.e., $\hat{I}_P(S) \geq (1 + \alpha)T$, where $\alpha$ is a user-defined parameter to guarantee that $\mathbb{E}[I_P(S)] \geq T$ holds with high probability. In the second stage, in a similar manner to the existing solution for IM [28], it iteratively selects the seed $u$ that leads to the largest $\hat{I}(u|S)$ until the budget is exhausted.

Let $S^o$ be the size-$k$ optimal solution for the IMP problem. In other words, $S^o$ is the size-$k$ seed set with the largest influence spread in $G$ while satisfying $\mathbb{E}[I_P(S^o)] \geq T$. On the theoretical side,

---

**Algorithm 1**: Select-And-Replace

| | |
|---|---|
| **Input** | : The graph $G$, the priority set $P$, the threshold $T$, the budget $k$ and the parameters $\gamma, \delta, \epsilon$. |
| **Output** | : The size-$k$ seed set $S$ |

1   $S \leftarrow$ SeedSelection$(G, P, T, k, \delta, \epsilon)$;      /* Select Stage */;
2   $\langle S, k_1^b \rangle \leftarrow$ GreedyReplace$(G, P, T, S, k, \gamma, \delta, \epsilon)$;    /* Replace Stage */;
3   **return** $\langle S, k_1^b \rangle$

---

IGS has the following theoretical result.

$$\Pr\left[\mathbb{E}[I(S)] \geq (1 - (1 - 1/k)^{k_2^a} - \epsilon)\mathbb{E}[I(S^o)]\right] \geq 1 - \delta, \quad (3)$$

where $k_2^a$ is the actual budget within the solution of IGS for maximizing the influence spread in $G$, i.e., $k_2^a$ is the number of seeds identified in the second stage of IGS. Clearly, the empirical approximation ratio of IGS is determined by $k_2^a$.

Limitations. Although IGS can return results efficiently, in some cases, it cannot provide any non-trivial approximation guarantee. For example, when $k = T = 100$ on the dataset Orkut with more than 100 million edges, IGS can only offer an approximation ratio of 0.034 (more details can be found in Section 5). This is because, when $T$ is large, IGS requires a significant portion of the budget to satisfy the threshold condition, leaving only a few budgets for the second stage, which results in a low empirical approximation ratio.

Moreover, IGS is inferior in maximizing the influence spread in $G$. In the IMP problem, to maximize the influence spread in $G$, it is optimal to maintain $\mathbb{E}[I_P(S)]$ as close to $T$ as possible, while ensuring the threshold condition is satisfied. Under such a setting, a larger fraction of the budget can remain available to maximize the total influence in $G$. However, IGS may deviate from this setting (i.e., $\mathbb{E}[I_P(S)]$ may far surpass $T$) due to the potential for the seeds selected in the second phase to influence the prioritized nodes.

## 3 Select-And-Replace Approach

In this section, we propose a novel framework, called Select-And-Replace (SAR), for the IMP problem, which consists of two stages. In Section 3.1, we first present an overview of SAR. Then, in Section 3.2 and Section 3.3, we introduce each stage within SAR in detail. Finally, we provide a theoretical analysis to demonstrate the correctness and expected time complexity of SAR in Section 3.4.

### 3.1 Framework Overview

The pseudocode of the SAR framework is shown in Algorithm 1. In general, SAR consists of two stages: *i)* select stage; *ii)* replace stage. In the first stage, our objective is to identify a size-$k$ seed set $S$ that maximizes the influence spread in $G$. This task is similar to the conventional IM problem but imposes a more stringent theoretical requirement to provide (data-dependent) approximation guarantees for the final solution of SAR. Specifically, in addition to ensuring that the returned solution $S$ provides $(1 - 1/e - \epsilon)$-approximation, we further ensure that its subset $S_j$ ($1 \leq j < k$), which consists of the seed nodes selected in the first $j$ iterations (under the greedy strategy), offers $(1 - (1 - 1/k)^j - \epsilon)$-approximation. Note that, the approximation guarantees for $S$ and $S_i$ are all established w.r.t. $S_{\text{IM}}^o$, which is the size-$k$ optimal solution that maximizes the influence spread in $G$. In the second stage, we iteratively replace the seed in $S$ until satisfying the threshold condition, i.e., $\mathbb{E}[I_P(S)] \geq T$.

---

**Algorithm 2:** SeedSelection

**Input** : The graph $G$, the priority set $P$, the threshold $T$, the budget $k$ and the parameters $\delta, \epsilon$.

**Output** : The size-$k$ seed set $S$

1   $\theta_{\max} \leftarrow \dfrac{2n\left((1-(1-1/k)^k)\sqrt{\ln \frac{6}{\delta}} + \sqrt{(1-(1-1/k)^k)(\ln\binom{n}{\lfloor n/2 \rfloor} + \ln \frac{6}{\delta})}\right)^2}{\epsilon^2 k}$;

2   $\theta_0 \leftarrow \theta_{\max} \cdot \epsilon^2 k/n$;

3   $i_{\max} \leftarrow \lceil \log_2 \frac{\theta_{\max}}{\theta_0} \rceil, a_1 \leftarrow \ln \frac{3 i_{\max}}{\delta}, a_2 \leftarrow \ln \frac{3 i_{\max}}{\delta}$;

4   generate two sets $\mathcal{R}_1, \mathcal{R}_2$ of $\theta_0$ random RR sets, respectively;

5   **for** $i \leftarrow 1$ *to* $i_{\max}$ **do**

6      identify a size-$k$ seed set $S \subseteq V$ using greedy strategy on $\mathcal{R}_1$, and record its all subsets $S_j$ for $1 \le j \le k$;

7      compute an upper bound $\sigma^U(S_{\text{IM}}^o)$ of $\mathbb{E}[I(S_{\text{IM}}^o)]$ based on $\mathcal{R}_1$;

8      Flag $\leftarrow 1$;

9      **for** $j \leftarrow 1$ *to* $k$ **do**

10         compute a lower bound $\sigma^L(S_j)$ of $\mathbb{E}[I(S_j)]$ based on $\mathcal{R}_2$;

11         **if** $\sigma^L(S_j)/\sigma^U(S_{\text{IM}}^o) < 1 - (1-1/k)^j - \epsilon$ **then**

12            Flag $\leftarrow 0$;

13            **break**;

14      **if** *Flag* $= 1$ *or* $i = i_{\max}$ **then**

15         **return** $S$;

16      double the sizes of $\mathcal{R}_1$ and $\mathcal{R}_2$ with new random RR sets;

---

**Algorithm 3:** GreedyReplace

**Input** : The graph $G$, the priority set $P$, the threshold $T$, the seed set returned in the first stage $S$, the budget $k$ and the parameters $\gamma, \delta, \epsilon$.

**Output** : The size-$k$ seed set $S$ and the integer $k_1^b$

1   $k_1^b \leftarrow k, \mathcal{R}_1^P \leftarrow \emptyset, \mathcal{R}_2^P \leftarrow \emptyset$;

2   $\mathcal{T}_2 \leftarrow 2(1+\gamma)\left(1+\frac{1}{3}\gamma\right)\ln\left(\frac{2\lceil(1+\gamma)T\rceil}{\delta}\right)\frac{1}{\gamma^2}$;

3   generate random PRR sets and store them into $\mathcal{R}_2^P$, until $Cov_{\mathcal{R}_2^P}(S) \ge \mathcal{T}_2$;

4   **if** $|P| \cdot Cov_{\mathcal{R}_2^P}(S)/|\mathcal{R}_2^P| \ge (1+\gamma)\cdot T$ **then**

5      **return** $\langle S, k \rangle$;

6   generate $|\mathcal{R}_2^P|$ random PRR sets and store them into $\mathcal{R}_1^P$;

7   **for** *each* $u \in S$ *with the reversing order of insertion* **do**

8      $S \leftarrow S \setminus \{u\}$;

9      $x \leftarrow \arg\max_{v \in V} Cov_{\mathcal{R}_1^P}(v|S)$;

10     $S \leftarrow S \cup \{x\}$;

11     $k_1^b \leftarrow k_1^b - 1$;

12     **while** $Cov_{\mathcal{R}_2^P}(S) < \mathcal{T}_2$ **do**

13        generate one random PRR set and store them into $\mathcal{R}_2^P$;

14     **if** $|P| \cdot Cov_{\mathcal{R}_2^P}(S)/|\mathcal{R}_2^P| \ge (1+\gamma)\cdot T$ **then**

15        **break**;

16   **return** $\langle S, k_1^b \rangle$;

---

Theoretically, SAR offers a $(1-(1-1/k)^{k_1^b} - \epsilon)$-approximation w.r.t. $S^o$ (Theorem 3.2), where $k_1^b$ is the actual budget within the solution of SAR that maximizes the influence spread in $G$.

### 3.2 Select Stage

To achieve the theoretical requirements stated above, we design a novel method by extending the state-of-the-art algorithm for IM (i.e., OPIM-C [28]), which only ensures the returned solution offers $(1 - 1/e - \epsilon)$-approximation ratio. As outlined in Algorithm 2, our method runs in an iterative manner. In each iteration, it initially generates two independent collections of RR sets, $\mathcal{R}_1$ and $\mathcal{R}_2$ (Line 4). The algorithm then employs a greedy approach on $\mathcal{R}_1$ to generate a solution $S$, that is, finding a set of $k$ nodes such that $S$ intersects with as many RR sets as possible in $\mathcal{R}_1$. Meanwhile, it records all the subsets $S_j$ of $S$ for $1 \le j \le k$ (Line 6). Subsequently, for any $j \in [1, k]$, $\sigma^L(S_j)$ and $\sigma^U(S_{\text{IM}}^o)$ are derived based on two concentration bounds (see Appendix A.1). In particular, $\sigma^L(S_j)$ is the lower bound of $\mathbb{E}[I(S_j)]$ and $\sigma^U(S_{\text{IM}}^o)$ is the upper bound of $\mathbb{E}[I(S_{\text{IM}}^o)]$, whose derivation can refer to the Theorem 4.2 and Theorem 4.3 in [28], and their expressions are shown below.

$$\sigma^L(S_j) = \left(\left(\sqrt{Cov_{\mathcal{R}_2}(S_j) + \frac{2a_1}{9}} - \sqrt{\frac{a_1}{2}}\right)^2 - \frac{a_1}{18}\right)\cdot \frac{n}{|\mathcal{R}_2|}, \quad (4)$$

$$\sigma^U(S_{\text{IM}}^o) = \left(\sqrt{\frac{Cov_{\mathcal{R}_1}(S)}{1-(1-1/k)^k} + \frac{a_2}{2}} + \sqrt{\frac{a_2}{2}}\right)^2 \cdot \frac{n}{|\mathcal{R}_1|}. \quad (5)$$

Then, based on the stopping condition in Line 11, the algorithm evaluates the quality of the generated solution $S$ and all of its subsets. If all the stopping conditions are satisfied (i.e., $\sigma^L(S_j)/\sigma^U(S_{\text{IM}}^o) \ge 1-(1-1/k)^j - \epsilon, \forall j \in [1, k]$) or $i = i_{\max}$, the solution $S$ is returned; otherwise, the sample size is doubled and aforementioned steps are repeated until the algorithm terminates.

### 3.3 Replace Stage

The first stage focuses solely on maximizing influence spread in $G$, without considering the threshold condition. To proceed, in the second stage, we replace some nodes in $S$ (the seed set returned in the first stage) to satisfy the threshold condition. Under this setting, we can ensure that $\mathbb{E}[I_P(S)]$ closely approximates $T$, which circumvents the limitation of IGS (i.e., it allocates an excessive budget to meet the threshold condition as discussed in Section 2.2), resulting in the improved influence spread.

As outlined in Algorithm 3, we first keep generating a set $\mathcal{R}_2^P$ of random PRR sets until the coverage $Cov_{\mathcal{R}_2^P}(S)$ of $S$ in $\mathcal{R}_2^P$ exceeds $\mathcal{T}_2$ (Lines 2-3), which ensures $\mathbb{E}[I_P(S)]$ can be estimated accurately via $\mathcal{R}_2^P$ on the theoretical side. If $|P| \cdot Cov_{\mathcal{R}_2^P}(S)/|\mathcal{R}_2^P|$ is not less than $(1+\gamma)\cdot T$, which means that the threshold condition has been satisfied currently, we directly return $S$ (Lines 4-5); otherwise, we come into the node replacement procedure (Lines 6-14). Generally, we consider processing the seeds in $S$ according to the reverse order of the insertion order. For each seed in $S$, it is first removed from $S$, and then we employ the greedy method on a newly generated set $\mathcal{R}_1^P$ of RR sets to identify the current best node and add it to $S$. Note that, we cannot directly identify the node based on $\mathcal{R}_2^P$, since using the same set of RR sets to both generate a seed set and estimate its influence spread will lead to biased estimation [16]. Since $Cov_{\mathcal{R}_2^P}(S)$ may decrease after each replacement, it is necessary to check whether the estimated value for $\mathbb{E}[I_P(S)]$ remains sufficiently accurate (Lines 12-13). Subsequently, if the updated seed set satisfies the threshold condition, we return $S$ as the final solution; otherwise, the replacement process continues.

### 3.4 Theoretical Analysis

In this section, we present a theoretical analysis for SAR. Specifically, we first show that the solution returned by the select stage, as well as its subsets, all provide reasonable approximation ratios

(Theorem 3.1). On this basis, we then show that SAR can provide a data-dependent theoretical guarantee, where the empirical approximation ratio is determined by the number of nodes replaced in the second stage (Theorem 3.2). Furthermore, we derive the expected time complexity for SAR (Theorem 3.3). Due to the limited space, the proofs for these theorems are omitted and can be found in Appendix A.3.

THEOREM 3.1. *Given* $0 \leq \epsilon, \delta \leq 1$, $S_{IM}^o$ *is the size-k optimal solution that maximizes the influence spread in G, SeedSelection (the first stage of SAR) returns* $S_j$ $(1 \leq j \leq k)$ *satisfies:*

$$\Pr\left[\mathbb{E}[I(S_j)] \geq (1 - (1 - 1/k)^j - \epsilon)\mathbb{E}[I(S_{IM}^o)]\right] \geq 1 - \delta. \quad (6)$$

THEOREM 3.2. *Given* $0 \leq \epsilon, \delta \leq 1$, $S^o$ *is the size-k optimal solution for IMP, SAR returns* $k_1^b$, $S$ *satisfies* $\Pr\left[\mathbb{E}[I_P(S)] \geq T\right] \geq 1 - \delta$ *and*

$$\Pr\left[\mathbb{E}[I(S)] \geq (1 - (1 - 1/k)^{k_1^b} - \epsilon)\mathbb{E}[I(S^o)]\right] \geq 1 - \delta. \quad (7)$$

THEOREM 3.3. *The expected time complexity of SAR is*

$$O\left(\frac{(\lfloor n/2 \rfloor \ln n + \ln(1/\delta))(m+n)}{\epsilon^2} + \frac{\mathbb{E}[I_P(v^*)] \cdot \ln(T/\delta) \cdot m}{\mathbb{E}[I_P(S)] \cdot \gamma^2}\right),$$

*where* $v^*$ *is selected randomly from those in G with probabilities proportional to their in-degrees.*

## 4 Adaptive-Alternation-Selection Approach

Compared to IGS, SAR demonstrates superior performance in terms of influence spread. In addition, the positive integer $k_1^b$ returned by SAR is always larger than $k_2^a$ returned by IGS, which implies that SAR consistently provides stronger approximation guarantees than IGS. More details can be found in Section 5.

Recall that IGS and SAR both employ the *non-adaptive* strategy, where the seeds are selected all at once, without any knowledge of the realization that would occur in the actual influence propagation process. Such a setting fails to take advantage of the previous spreading results when selecting the next seed node, which may lead to unsatisfactory results. Moreover, the selected $S$ may influence fewer than $T$ prioritized nodes for some realizations or much more than $T$ prioritized nodes for some other realizations, both of which are undesirable scenarios. For example, SAR may return $S$ that fails to influence $T$ prioritized nodes, since the guarantee $\mathbb{E}[I_P(S)] \geq T$ is subject to $\delta$ probability of failure (Theorem 3.2).

To obtain more practical results, in this section, we further study the IMP problem in the *adaptive* setting, which has been shown to be more effective than the non-adaptive strategy in many real-world applications [1, 7, 8]. In a nutshell, the general idea of the adaptive strategy is to iteratively select seed nodes based on the observed diffusion results of the previously chosen seeds. Under such a setting, we design an effective algorithm named AAS with an expected approximation. To facilitate understanding, we first introduce some frequently used notations in Section 4.1. Subsequently, in Section 4.2 and Section 4.3, we thoroughly present the framework and the theoretical guarantees it provides, respectively.

### 4.1 Useful Notations

**Partial realization**. Given a realization $\phi$, let $\phi(v)$ be the activation state of $v$ under $\phi$, i.e., the statuses (either live or blocked) of all edges that would be explored after activating $v$. Next, we introduce the concept of *partial realization* $\psi$, which presents the observation that we have made so far. Let $\text{dom}(\psi)$ be the domain of $\psi$, i.e., the set of nodes that have been observed. Similarly, $\psi(v)$ is the activation state of $v$ under $\psi$. We say $\psi$ is *consistent* with $\phi$, denoted to $\phi \sim \psi$, if for every $v \in \text{dom}(\psi), \psi(v) = \phi(v)$. Furthermore, we introduce the concept of *residual graph*. Given a partial realization $\psi$, a residual graph is the subgraph of $G$ constructed by removing all activated nodes in $\psi$ and their incident edges from $G$.

**Policy**. Under the adaptive setting, a *policy* $\pi$ is the strategy for selecting the next node based on current partial realization $\psi$. Given a fixed $\psi$, if $\pi$ always selects the same seed node, we say this policy is *deterministic*; otherwise, it is *randomized*. We use $\omega$ to denote all possible randomness brought by the randomized policy and $\pi(\omega)$ be a policy w.r.t. $\omega$. Let $\pi(\omega, \psi)$ be the selected node by $\pi(\omega)$ under $\psi$, and $\mathcal{E}(\pi(\omega), \phi)$ be the set of selected nodes by $\pi(\omega)$ under $\phi$. Accordingly, the influence spread of policy $\pi(\omega)$ is defined below.

$$\sigma(\pi(\omega)) = \mathbb{E}[I(\pi(\omega))] = \mathbb{E}_\Phi[I_\Phi(\mathcal{E}(\pi(\omega), \Phi))].$$

Additionally, for any $\psi$, let $\Delta(v|\psi)$ and $\Delta(\pi(\omega)|\psi)$ denote the *conditional marginal benefit* of the node $v$ and a policy $\pi(\omega)$ conditioned on $\psi$, respectively. The formal definitions are shown below.

$$\Delta(v|\psi) = \mathbb{E}_{\Phi \sim \psi}[I_\Phi(\text{dom}(\psi) \cup \{v\}) - I_\Phi(\text{dom}(\psi))],$$
$$\Delta(\pi(\omega)|\psi) = \mathbb{E}_{\Phi \sim \psi}[I_\Phi(\text{dom}(\psi) \cup \mathcal{E}(\pi(\omega), \Phi)) - I_\Phi(\text{dom}(\psi))].$$

### 4.2 General Framework

To solve adaptive IMP, a straightforward idea is to extend the SAR to the adaptive setting. However, it is infeasible for us to implement the replacement procedure of SAR in an adaptive manner. This is because, in the adaptive setting, selecting the next seed is based on the actual propagation information. In other words, before we select the next seed, the propagation of the previously selected seeds has been finished. Therefore, we cannot regret the selection of a node and then choose another one with the adaptive strategy.

**AAS approach**. To design an effective method with the adaptive strategy, the key lies in appropriately allocating the budgets for maximizing the influence spread in $G$ and in $P$. For this purpose, an ideal approach is to first select $k_1$ nodes to maximize $\mathbb{E}[I(\cdot)]$, followed by utilizing the remaining $k - k_1$ nodes to meet the threshold condition, ensuring that the condition is satisfied just when the budget is fully expended. However, this approach is impractical, as the value of $k_1$ cannot be known in advance. To address this issue, we propose a general framework named Adaptive-Alternation-Selection (AAS), which could automatically allocate the budgets for the two tasks. The rationale behind the automation mechanism is: in the $i$-th iteration, AAS will select a seed that maximizes $\mathbb{E}[I(\cdot)]$ if the remaining budget (i.e., $k - i$) is still sufficient to meet the threshold condition. Otherwise, it will select the seed that maximizes $\mathbb{E}[I_P(\cdot)]$.

The detailed pseudocode of AAS is shown in Algorithm 4. In a nutshell, AAS consists of $k$ iterations. In each iteration, there are two procedures that have the potential to be implemented. One is shown in Lines 4-6, whose objective is to identify a node $u_i$ from $G_i$ such that in expectation (w.r.t $\omega$) $u_i$ has an influence spread at least $\alpha$ times that of the optimal node on the basis of the $\psi_{i-1}$, which is the partial realization after selecting the first $i - 1$ nodes. For this purpose, we could directly invoke the algorithm EptAIM [14]

---

**Algorithm 4**: Adaptive-Alternation-Selection

**Input** : The graph $G$, the priority set $P$, the threshold $T$, the budget $k$, the sample size for PRR set $\theta_p$ and the parameter $\alpha$.

**Output** : The size-$k$ seed set $S$ and the integer $k^c$

1  $G_1 \leftarrow G, S \leftarrow \emptyset, I_P(S) \leftarrow 0, k^c \leftarrow 0$;

2  **for** $i \leftarrow 1$ $to$ $k$ **do**

3      **if** $T - I_P(S) \leq k - |S| - 1$ **then**

4          select $u_i$ from $G_i$ that $\mathbb{E}_\omega[\Delta(u_i|\psi_{i-1})] \geq \alpha \cdot \max_{u \in V} \Delta(u|\psi_{i-1})$;

5          $S \leftarrow S \cup \{u_i\}$;

6          $k^c \leftarrow k^c + 1$;

7      **else**

8          clear all the PRR sets in $\mathcal{R}^P$, generate $\theta_p$ PRR sets based on $G_i$ and store them into $\mathcal{R}^P$;

9          $u_i^P \leftarrow \arg\max_{v \in V} Cov_{\mathcal{R}^P}(v|S)$;

10         $S \leftarrow S \cup \{u_i^P\}$;

11      Observe the influence of $u_i$ (or $u_i^P$) in $G_i$ and increase $I_P(S)$ accordingly;

12      Remove the activated nodes from $G_i$ and obtain the residual graph $G_{i+1}$;

13  **return** $\langle S, k^c \rangle$

---

(with the setting $k = 1$), which is the state-of-the-art method for adaptive IM and offers an expected approximation ratio. The other procedure is presented in Lines 8-10, with the goal of selecting a node $u_i^P$ that maximizes the influence spread in $P$. Here we employ a heuristic to accelerate the process, i.e., by fixing the sample size of $\mathcal{R}^P$ to a constant and applying a greedy method on $\mathcal{R}^P$. Despite this, AAS still provides the non-trivial approximation guarantee (Theorem 4.2).

Based on a judgment condition (Line 3), AAS smartly chooses one procedure to execute in each iteration. Subsequently, it observes the newly activated nodes and updates the corresponding information (Lines 11-12). Note that, $I_P(S)$ is the actual influence spread of $S$ (i.e., the number of activated nodes) in $P$ based on the observation we have made so far. The process stops until the budget is exhausted.

### 4.3 Theoretical Analysis

Next, we will provide a detailed theoretical analysis for AAS. In particular, we first introduce a critical lemma that establishes a relationship between $\pi$ and $\pi^I$ (Lemma 4.1), where $\pi^I$ is a randomized policy with $k^c$ iterations for the task of maximizing the influence spread in $G$ and $k^c$ is the actual budget within the solution of AAS that maximizes the influence spread in $G$. On the basis of this lemma, we show that AAS offers a $(1 - e^{(\epsilon-1) \cdot k^c/k})$-expected approximation (Theorem 4.2). Then, we demonstrate that AAS ensures $I_P(S) \geq T$ holds in all instances (Theorem 4.3), making it more effective than SAR. Finally, the expected time complexity of AAS is derived (Theorem 4.4). Due to space constraints, we omit all the proofs here, which are available in the Appendix A.4.

LEMMA 4.1. *For any realization $\phi$, we have*

$$I_\phi(\mathcal{E}(\pi(\omega), \phi)) \geq I_\phi(\mathcal{E}(\pi^I(\omega), \phi)). \quad (8)$$

THEOREM 4.2. *$\pi(\omega)$ be the policy employed by AAS. For any policy $\pi^*(\omega)$ that satisfies the threshold requirement, we have*

$$\mathbb{E}_\omega[\sigma(\pi(\omega))] \geq (1 - e^{\frac{(\epsilon-1) \cdot k^c}{k}}) \cdot \mathbb{E}_\omega[\sigma(\pi^*(\omega))]. \quad (9)$$

THEOREM 4.3. *AAS returns a solution $S$ such that $I_P(S) \geq T$ holds in all instances.*

**Table 1: Statistics of datasets**

| Dataset | Type | $|V|$ | $|E|$ | Avg. deg |
|---------|------|-------|-------|----------|
| NetHEPT | undirected | 15,229 | 31,376 | 4.18 |
| DBLP | undirected | 317,080 | 1,049,866 | 6.60 |
| Twitter | directed | 81,306 | 1,768,149 | 59.5 |
| Youtube | undirected | 1,134,890 | 2,987,624 | 5.30 |
| Orkut | undirected | 3,072,441 | 117,185,083 | 76.3 |

THEOREM 4.4. *The expected time complexity of AAS is*

$$O\left(k \cdot \max\{(\log n + \log \frac{1}{\epsilon})(m+n)/\epsilon^2, \frac{\mathbb{E}[I_P(v^*)]}{|P|}m\theta_p\}\right),$$

*where $v^*$ is selected randomly from those in $G$ with probabilities proportional to their in-degrees.*

## 5 Experiments

In this section, we conduct extensive experiments on 5 real-world datasets to evaluate the performance of our methods.

### 5.1 Experimental Settings

**Algorithms**. In the experiment, we implement the following three algorithms. *i)* **IGS**: the state-of-the-art algorithm for the IMP problem proposed in [24] (details can be found in Section 2.2); *ii)* **SAR**: the algorithm proposed in Section 3; *iii)* **AAS**: the algorithm proposed in Section 4. In addition, we incorporate the incremental update technique [12] into AAS to further accelerate the algorithm.

**Datasets**. We use 5 real datasets which are available on SNAP[1] in our experiments. The details of the datasets are presented in Table 1. For each dataset, we select the top-200 nodes with the highest in-degrees and store them into the priority set.

**Parameter settings**. Following the convention [28–30], we set the propagation probability $p(u, v)$ of each edge $\langle u, v \rangle$ as the inverse of $v$'s in-degree. By default, for the non-adaptive algorithms IGS and SAR, we set $\epsilon = \gamma = 0.1$, $\delta = 1/n$, and we repeat each algorithm 20 times to report the average value. For the adaptive method AAS, we set $\theta_p = 10000$, and following [12, 14], we adopt a more relaxed deviation parameter, i.e., $\epsilon = 0.5$, as a tighter approximation guarantee would incur the much higher computational cost. In addition, we generate 20 random realizations to report the average influence spread. We evaluate the performance of our algorithms according to the $k$-setting and $T$-setting. Under the $k$-setting, we fix $T = 100$ and vary $k$ such that $k \in \{100, 120, 140, 160, 180\}$. Under the $T$-setting, we fix $k = 200$ and vary $T$ such that $T \in \{100, 120, 140, 160, 180\}$. All the programs are implemented in C++ and performed on a PC with an Intel Xeon 2.10GHz CPU and 256GB memory.

### 5.2 Experimental Performance

**Influence spread**. We first evaluate the performance of influence spread for each algorithm. The experimental results under the $k$-setting and $T$-setting are reported in Figure 1. As shown, SAR and AAS consistently outperform IGS in terms of influence spread on all

---

[1]http://snap.stanford.edu

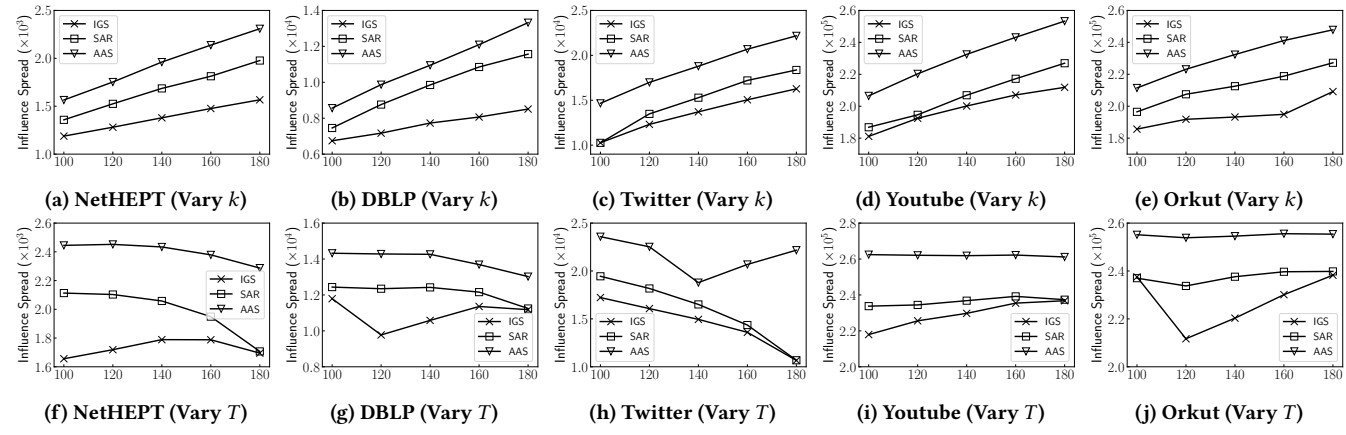

**Figure 1: Influence spread evaluation by varying $k$ and $T$**

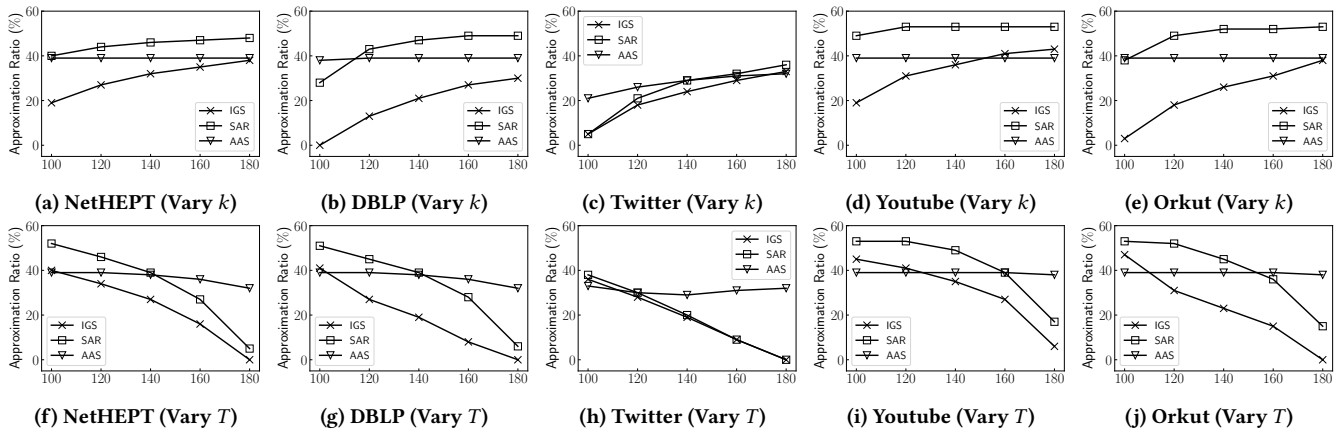

**Figure 2: Approximation ratio evaluation by varying $k$ and $T$**

the datasets. In particular, compared to IGS, SAR achieves around 22.3% larger spread, and AAS achieves around 42.6% larger spread on NetHEPT. The main reason for this observation is that compared to SAR and AAS, IGS requires a larger portion of the budget to satisfy the threshold condition, thereby leaving less budget available for maximizing influence spread in $G$. In addition, AAS always provides better performance compared to SAR. This is because AAS takes advantage of the previous spreading results when selecting the next seed node, thereby preventing a node from being activated multiple times and ultimately enhancing the influence spread.

Moreover, as can be seen in Figures 1(a)-1(e), the influence of all algorithms increases as $k$ increases, which is not surprising given the larger budget available for initiating influence propagation. Besides, as shown in Figures 1(f)-1(j), the influence spread of three algorithms generally exhibits a declining trend as $T$ increases. The primary reason is that with the increase of $T$, a larger budget is required to satisfy the threshold condition, consequently leading to a reduction in the budget for maximizing the influence spread in $G$.

**Approximation ratio**. Then, we calculate and report the empirical approximation ratios for all three algorithms based on Eq. (3), Eq. (7) and Eq. (9), respectively. As can be seen in Figure 2, SAR consistently

outperforms IGS in terms of approximation ratio on all the datasets. In particular, on Orkut with the setting $k = T = 100$, IGS almost cannot provide any theoretical guarantee while SAR can provide strong theoretical guarantee, achieving a ratio of approximately 40%. This is because SAR first identifies the most influential nodes in graph $G$. These nodes also have the potential to activate nodes in $P$. In some cases, the threshold condition is met after completing the first phase of SAR, which allows it to directly return the seed nodes and leads to strong theoretical guarantee. In contrast, IGS initially attempts to meet the threshold condition, which may consume a significant portion of the budget when $T$ is large. This leaves fewer budgets for selecting nodes in the second phase, resulting in a lower approximation ratio. Additionally, in most cases, AAS achieves a higher approximation ratio than IGS, and in a few cases, AAS outperforms SAR. The reason that AAS occasionally provides inferior performance than IGS and SAR is that we set the deviation parameter $\epsilon$ of AAS as 0.5. More specifically, there is a trade-off between the approximation ratio and efficiency. If we set $\epsilon = 0.1$ as done with IGS and SAR, the empirical approximation ratio of AAS can be significantly improved, however, the side effect is that AAS will encounter scalability issues.

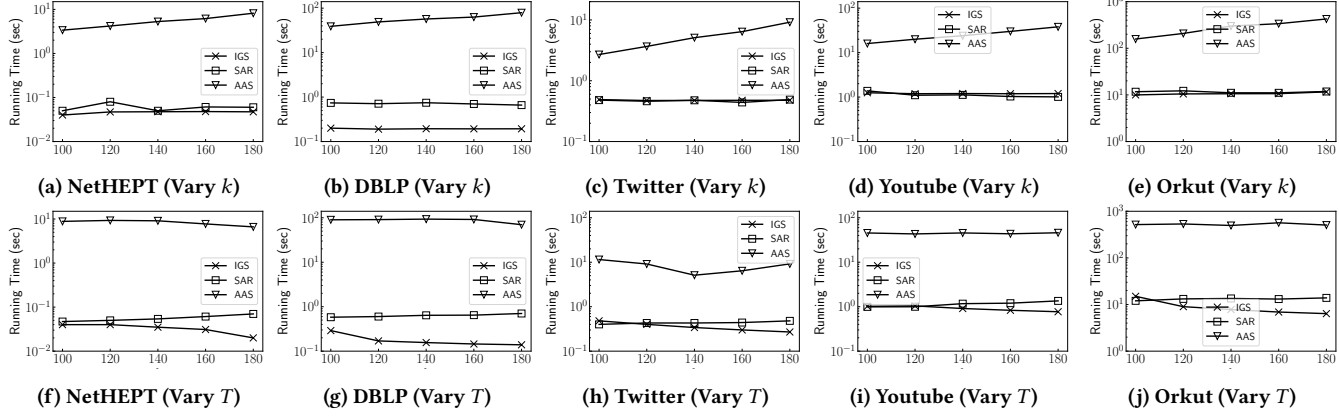

**Figure 3: Running time evaluation by varying $k$ and $T$**

Furthermore, as $k$ and $T$ increase, the trend of the approximation ratio mirrors that of the influence spread. This is because the approximation ratios of the three methods are all primarily driven by the budget for maximizing the total influence in $G$.

**Running time**. Further, we evaluate the running time for each algorithm, and the results are demonstrated in Figure 3. As can be seen, the time cost of IGS and SAR is quite similar, and both exhibit high efficiency. In contrast, AAS takes much more time than IGS and SAR. It can be explained that AAS requires generating a significantly larger number of samples, which naturally results in increased running time. Nevertheless, AAS can still easily scale to the large graphs. For example, AAS only requires around 500 seconds to complete when handling Orkut with more than one hundred million edges.

## 6  Related Work

**Non-adaptive IM**. Kempe et al. [18] first formulate the influence maximization (IM) problem, which aims to identify a set of seeds with the largest influence spread. They accordingly introduce two basic propagation models, i.e., *independent cascade* (IC) and *linear threshold* (LT) models. To address the problem, they utilize a polynomial-time greedy algorithm that returns $(1 - 1/e - \epsilon)$-approximate solution. More specifically, a Monte-Carlo based approach is leveraged for estimating the influence spread of any seed set $S$. Afterwards, a large number of work focuses on developing heuristic algorithms to reduce computational overhead [4–6, 11, 17, 23, 34]. However, as a side effect, these solutions yield results without theoretical guarantees. Brogs et al. [2] make a theoretical breakthrough by proposing the elegant *Reverse Influence Sampling* (RIS) technique, which reduces the time complexity to almost linear to the graph size. Subsequently, many RIS-based algorithms [16, 22, 28–30] have been proposed, which ensure $(1 - 1/e - \epsilon)$-approximations while reducing computational overhead. Besides, a plethora of research work focuses on more practical scenarios rather than the classic IM, such as incorporating the time aspect [19, 21] and location aspect [33]. Furthermore, inspired by real marketing scenarios in social networks, priority-aware IM is proposed [24], which is the focus of this paper. However, existing methods have notable limitations in terms of effectiveness, motivating us to develop novel and improved approaches.

**Adaptive IM**. Golovin et al. [10] first introduce the concept of adaptive submodularity, where the seed selection is based on the observation of previous diffusion results, and prove that the adaptive greedy policy can provide a $(1 - 1/e)$-approximate solution. Subsequently, Han et al. [13] and Sun et al. [26] propose two algorithms for adaptive IM, and their algorithms are claimed to provide the same worst-case approximation guarantee of $1 - e^{(1-1/e)(\epsilon-1)}$ with high probability. However, Huang et al. [14] point out deficiencies in their proofs. To tackle this issue, they design a novel framework for adaptive IM with $(1 - e^{(1-(1-1/b)^b)(\epsilon-1)})$-expected approximation. Recently, Guo et al. [12] study the budgeted adaptive IM problem, which is the adaptive IM problem under the skewed cost model (i.e., the cost of each node may not be equal). They propose a practical algorithm that provides the expected approximation guarantee. In addition, they devise an incremental update approach, which can be easily extended to the adaptive IM problem to improve efficiency. Furthermore, several variants of the adaptive IM problem [15, 27, 35] are studied, all of which are based on the adaptive framework that utilizes feedback from previous selections to enable more accurate node selection.

## 7  Conclusion

In this paper, we study the influence maximization with priority problem. To begin with, we revisit the state-of-the-art methods for IMP and point out their limitations. To fill the gap, we propose a novel algorithm SAR with both superior empirical effectiveness and strong theoretical guarantees. Besides, to obtain more practical results, we conduct the first research to study IMP under the adaptive setting, where the seeds are iteratively selected after observing the diffusion result of the previous seeds. We design a scalable and effective algorithm AAS that achieves expected approximation guarantees. Finally, comprehensive experiments on 5 real-world datasets are conducted to validate the performance of SAR and AAS. The experimental results show that SAR and AAS can demonstrate better effectiveness and offer a higher empirical approximation ratio compared to the state-of-the-art method.

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

# A Appendix

## A.1 Concentration Bounds

Lemma A.1 ([29]). *Given a seed set $S$ and a fixed number of $\theta$ random RR sets $\mathcal{R}$. For any $\lambda > 0$,*

$$\Pr\left[Cov_{\mathcal{R}}(S) - \theta \cdot \frac{\mathbb{E}[I(S)]}{n} \geq \lambda\right] \leq \exp\left(\frac{-\lambda^2}{\frac{2\lambda}{3} + \frac{2\mathbb{E}[I(S)]\theta}{n}}\right), \quad (10)$$

$$\Pr\left[Cov_{\mathcal{R}}(S) - \theta \cdot \frac{\mathbb{E}[I(S)]}{n} \leq -\lambda\right] \leq \exp\left(\frac{-\lambda^2}{2\theta \cdot \frac{\mathbb{E}[I(S)]}{n}}\right). \quad (11)$$

## A.2 Proofs for Results in Section 2

**Proof of Lemma 2.1.** According to [18], the IM problem is proved to be NP-hard and cannot be approximated within a ratio of $1 - 1/e + \epsilon$ for any $\epsilon > 0$ unless P = NP. Since IMP will degenerate to the IM problem when $P = \emptyset$, the lemma follows. □

## A.3 Proofs for Results in Section 3

**Proof of Theorem 3.1.** SeedSelection returns the solution in two scenarios: $i = i_{\max}$ or $i < i_{\max}$. We first consider the case of $i = i_{\max}$, where satisfies $|\mathcal{R}_1| = |\mathcal{R}_2| = \theta \geq \theta_{\max}$. Let $\theta_1 = \frac{2n\ln(6/\delta)}{\epsilon_1^2 \cdot \mathbb{E}[I(S_{\text{IM}}^o)]}$ and $\epsilon_1 < \epsilon$. When $\theta \geq \theta_1$, we have:

$$\Pr\left[n \cdot \frac{Cov_{\mathcal{R}_1}(S_{\text{IM}}^o)}{\theta} \leq (1 - \epsilon_1)\mathbb{E}[I(S_{\text{IM}}^o)]\right]$$

$$\leq \exp\left[-\frac{\epsilon_1^2}{2} \cdot \theta \cdot \frac{\mathbb{E}[I(S_{\text{IM}}^o)]}{n}\right] \leq \frac{\delta}{6}, \quad (12)$$

where the first inequality is due to Eq. (11). Based on the submodularity and monotonicity of $Cov_{\mathcal{R}_1}(\cdot)$, it is easy to deduce

$$n \cdot \frac{Cov_{\mathcal{R}_1}(S_j)}{\theta} \geq \left(1 - (1 - \frac{1}{k})^j\right) \cdot n\frac{Cov_{\mathcal{R}_1}(S_{\text{IM}}^o)}{\theta}. \quad (13)$$

Combining Eq. (12) and Eq. (13), we have

$$\Pr\left[n \cdot \frac{Cov_{\mathcal{R}_1}(S_j)}{\theta} \geq \left(1 - (1 - \frac{1}{k})^j\right)(1 - \epsilon_1)\mathbb{E}[I(S_{\text{IM}}^o)]\right] \geq 1 - \frac{\delta}{6}. \quad (14)$$

Let $\theta_2^j = \frac{(2 - 2(1 - \frac{1}{k})^j)n\ln(\binom{n}{j}\cdot 6/\delta)}{\mathbb{E}[I(S_{\text{IM}}^o)]\epsilon_2^2}$ and $\epsilon_2 = \epsilon - (1 - (1 - \frac{1}{k})^j) \cdot \epsilon_1$, when $\theta \geq \theta_2^j$, assume that $\mathbb{E}[I(S_j)] < (1 - (1 - 1/k)^j - \epsilon)\mathbb{E}[I(S_{\text{IM}}^o)]$ holds, we have

$$\Pr[n \cdot \frac{Cov_{\mathcal{R}_1}(S_j)}{\theta} - \mathbb{E}[I(S_j)] \geq \epsilon_2\mathbb{E}[I(S_{\text{IM}}^o)]]$$

$$\leq \exp\left(-\frac{\epsilon_2^2 \cdot \mathbb{E}[I(S_{\text{IM}}^o)]^2 \cdot \theta}{(2\mathbb{E}[I(S_j)] + \frac{2}{3}\mathbb{E}[I(S_{\text{IM}}^o)] \cdot \epsilon_2) \cdot n}\right)$$

$$\leq \exp\left(-\frac{\epsilon_2^2 \cdot \mathbb{E}[I(S_{\text{IM}}^o)] \cdot \theta}{(2(1 - (1 - \frac{1}{k})^j - \epsilon) + \frac{2}{3}\epsilon_2) \cdot n}\right) \leq \frac{\delta}{6 \cdot \binom{n}{j}}, \quad (15)$$

where the first inequality is due to Eq. (10). According to Eq. (14), Eq. (15) and there exists at most $\binom{n}{j}$ seed sets, when $\theta \geq \max\{\theta_1, \theta_2^j\}$, we have $\mathbb{E}[I(S_j)] \geq (1 - (1 - 1/k)^j - \epsilon)\mathbb{E}[I(S_{\text{IM}}^o)]$ holds with at least $1 - \frac{\delta}{3}$ probability, which contradicts the assumption. Then by setting $\theta_1 = \theta_2^j$, we can deduce a sample size threshold $\theta_T^j$ such that

$\theta_T^j \geq \max\{\theta_1, \theta_2^j\}$ for each $j \in [1, k]$.

$$\theta_T^j = \frac{2n\left((1 - (1 - 1/k)^j)\sqrt{\ln\frac{6}{\delta}} + \sqrt{(1 - (1 - 1/k)^j)(\ln\binom{n}{j} + \ln\frac{6}{\delta})}\right)^2}{\epsilon^2 k}. \quad$$

Recall that $\theta_{\max}$ is defined in Line 1 of Algorithm 2. It is easy verify that $\forall j \in [1, k], \theta_{\max} \geq \theta_T^j$. Therefore, we can conclude that when $\theta \geq \theta_{\max}, \forall j \in [1, k]$,

$$\Pr[\mathbb{E}[I(S_j)] \geq (1 - (1 - 1/k)^j - \epsilon)\mathbb{E}[I(S_{\text{IM}}^o)]] \geq 1 - \frac{\delta}{3}. \quad (16)$$

Next, we consider the case when $i < i_{\max}$. According to [28], for $\forall j \in [1, k]$, we have $\Pr[\sigma^L(S_j) \leq \mathbb{E}[I(S_j)]] \geq 1 - \frac{\delta}{3i_{\max}}$ and $\Pr[\sigma^U(S_{\text{IM}}^o) \geq \mathbb{E}[I(S_{\text{IM}}^o)]] \geq 1 - \frac{\delta}{3i_{\max}}$ in each of the first $i_{\max} - 1$ iterations. If $\sigma^L(S_j)/\sigma^U(S_{\text{IM}}^o) \geq 1 - (1 - 1/k)^j - \epsilon$, then with at least $1 - 2\delta/3i_{\max}$ probability, $\mathbb{E}[I(S_j)]/\mathbb{E}[I(S_{\text{IM}}^o)] \geq 1 - (1 - 1/k)^j - \epsilon$ holds. By the union bound, for any $1 \leq j \leq k$, $S_j$ is a $(1 - (1 - 1/k)^j - \epsilon)$-approximate solution with at least $1 - 2\delta/3$ probability in the first $i_{\max} - 1$ iterations. Therefore, the theorem holds. □

**Proof of Theorem 3.2.** Based on the result of Theorem 3.1, $\mathbb{E}[I(S)] \geq \mathbb{E}[I(S_{k_1^b})]$ and $\mathbb{E}[I(S_{\text{IM}}^o)] \geq \mathbb{E}[I(S^o)]$, we have

$$\Pr\left[\mathbb{E}[I(S)] \geq (1 - (1 - 1/k)^{k_1^b} - \epsilon)\mathbb{E}[I(S^o)]\right] \geq 1 - \delta. \quad (17)$$

Then, according to the result derived by Zhu et al. [36], that is, when $Cov_{\mathcal{R}_2^P}(S) \geq \mathcal{T}_2$, $\frac{|P|}{|\mathcal{R}_2^P|} \cdot Cov_{\mathcal{R}_2^P}(S)$ is a $(\gamma, \frac{\delta}{\lceil(1+\gamma)T\rceil})$-estimate of $\mathbb{E}[I_P(S)]$, i.e.,

$$\Pr\left[(1 - \gamma)\mathbb{E}[I_P(S)] \leq \frac{|P|}{|\mathcal{R}_2^P|} \cdot Cov_{\mathcal{R}_2^P}(S) \leq (1 + \gamma)\mathbb{E}[I_P(S)]\right] \geq 1 - \frac{\delta}{\lceil(1+\gamma)T\rceil}.$$

After each replacement, we ensure that the value of $Cov_{\mathcal{R}_2^P}(S)$ remains greater than $\mathcal{T}_2$. In addition, there are at most $\lceil(1 + \gamma)T\rceil$ iterations. Thus, by the union bound, there is at least $1 - \delta$ probability that $\frac{|P|}{|\mathcal{R}_2^P|} \cdot Cov_{\mathcal{R}_2^P}(S) \leq (1 + \gamma)\mathbb{E}[I_P(S)]$ holds for all iterations of GreedyReplace. Based on this observation, it is trivial to verify that when $|P| \cdot Cov_{\mathcal{R}_2^P}(S)/|\mathcal{R}_2^P| \geq (1 + \gamma) \cdot T, \Pr[\mathbb{E}[I_P(S)] \geq T] \geq 1 - \delta$. The theorem holds. □

**Proof of Theorem 3.3.** We consider the expected time complexity required for the two stages separately. For the first stage, since the framework is similar to OPIM-C [28], and the sample size is upper bounded by $\theta_{\max}$, thus, it is easy to derive that the time complexity of the first stage is $O(\frac{(\lfloor n/2 \rfloor \ln n + \ln(1/\delta))(m+n)}{\epsilon^2})$.

For the second stage, based on the stopping condition for the PRR set generation (i.e., $Cov_{\mathcal{R}_2^P}(S) \geq \mathcal{T}_2$), we can deduce that the expected number of PRR sets generated in the second stage is $O(\mathcal{T}_2 \cdot \frac{|P|}{\mathbb{E}[I_P(S)]})$. The expected time required to generate a PRR set (denoted by $ETP$) is $\frac{\mathbb{E}[I_P(v^*)]}{|P|} \cdot m$. To explain, let $p_\phi$ be the probability that a randomly selected edge from $E$ ends at a node in $R_\phi^P$, which is a PRR set determined by $\phi$. Then we have $ETP = \mathbb{E}_{\Phi \sim \Omega}[p_\Phi \cdot m]$. Let $ID$ be the probability distribution over the nodes in $G$, such that the probability mass for each node is proportional to its in-degree in $G$. Let $v^*$ be a node sampled from $ID$ and $\Pr[v^* \sim ID]$ be the corresponding probability. $\mathcal{I}(v^*, R_\phi^P)$ be an indicator variable that

equals 1 if $v^* \in R_\phi^P$, and 0 otherwise. Then for any $\phi$, we have $p_\phi = \sum_{v^*} \Pr[v^* \sim ID] \cdot \mathcal{I}(v^*, R_\phi^P)$. We have:

$$
\begin{aligned}
\frac{ETP}{m} = \mathbb{E}_\Phi[p_\Phi] &= \sum_{\phi \in \Omega} p(\phi) \cdot p_\phi \\
&= \sum_{\phi \in \Omega} p(\phi) \cdot \sum_{v^*} \Pr[v^* \sim ID] \cdot \mathcal{I}(v^*, R_\phi^P) \\
&= \sum_{v^*} \Pr[v^* \sim ID] \cdot \sum_{\phi \in \Omega} p(\phi) \cdot \mathcal{I}(v^*, R_\phi^P) = \frac{\mathbb{E}[I_P(v^*)]}{|P|}.
\end{aligned}
$$

Based on Wald's equation [31], the expected time cost of the second stage is $O\left(\frac{\mathbb{E}[I_P(v^*)] \cdot \ln(T/\delta) \cdot m}{\mathbb{E}[I_P(S)] \cdot \gamma^2}\right)$. To sum up, the theorem holds. □

## A.4 Proofs for Results in Section 4

**PROOF OF THEOREM 4.1.** Without loss of generality, for any realization $\phi$, assume that $\mathcal{E}(\pi^I(\omega), \phi) = \{v_{k^c}^1, v_{k^c}^2, \ldots, v_{k^c}^{k^c}\}$ and $\mathcal{E}(\pi(\omega), \phi) = \{x_{k_b}^1, u_{k^c}^1, x_{k_b}^2, u_{k^c}^2, \ldots, x_{k_b}^{k_b}, u_{k^c}^n, \ldots, u_{k^c}^{k^c}\}$, where $k_b = k - k^c$ and $x_{k_b}^i$ (resp. $u_{k^c}^i$) is the $i$-th selected node for maximizing the influence spread in $P$ (resp. $G$). Define a new randomized policy $\pi'$, with $\mathcal{E}(\pi'(\omega), \phi) = \{x_{k_b}^1, v_{k^c}^1, x_{k_b}^2, v_{k^c}^2, \ldots, x_{k_b}^{k_b}, v_{k^c}^n, \ldots, v_{k^c}^{k^c}\}$. We have

$$I_\phi(\mathcal{E}(\pi(\omega), \phi)) \geq I_\phi(\mathcal{E}(\pi'(\omega), \phi)), \tag{18}$$

since $u_{k^c}^i$ always contributes the largest marginal gain for $I_\phi(\cdot)$ in the current iteration. In addition, based on the monotonicity of $I_\phi(\cdot)$ and $\mathcal{E}(\pi^I(\omega), \phi) \subseteq \mathcal{E}(\pi'(\omega), \phi)$, we have

$$I_\phi(\mathcal{E}(\pi'(\omega), \phi)) \geq I_\phi(\mathcal{E}(\pi^I(\omega), \phi)). \tag{19}$$

According to Eq. (18) and Eq. (19), the lemma holds. □

**PROOF OF THEOREM 4.2.** Based on Lemma 4.1, for any $\phi$ and $\omega$, we have $I_\phi(\mathcal{E}(\pi(\omega), \phi) \geq I_\phi(\mathcal{E}(\pi^I(\omega), \phi)$ holds. By taking the expectation over the randomness of $\omega$ and $\phi$, we have

$$\mathbb{E}_\omega[\sigma(\pi(\omega))] \geq \mathbb{E}_\omega[\sigma(\pi^I(\omega))]. \tag{20}$$

Let $\pi'$ be any randomized policy with $k$ iterations for influence maximization, and $\pi_i$ be the policy that performs exactly the same as $\pi$, except that $\pi_i$ only selects the first $i$ nodes for any $i \leq k$. According to [14], the following equation holds.

$$\mathbb{E}_\omega[\sigma(\pi'(\omega)) - \sigma(\pi_{k^c}^I(\omega))] \leq (1 - \frac{\alpha}{k}) \cdot \mathbb{E}_\omega[\sigma(\pi'(\omega)) - \sigma(\pi_{k^c-1}^I(\omega))],$$

where $\alpha$ is the expected approximation ratio of EptAIM [14]. Under our setting (i.e., in each iteration we only select one node), $\alpha$ equals $1 - \epsilon$. Besides, for any $x$ such that $0 \leq x \leq 1$, we have $1 - x \leq e^{-x}$. Therefore, by the recursive calculation, we have

$$
\begin{aligned}
&\mathbb{E}_\omega[\sigma(\pi'(\omega)) - \sigma(\pi_{k^c}^I(\omega))] \\
\leq &e^{-\frac{1-\epsilon}{k}} \cdot \mathbb{E}_\omega[\sigma(\pi'(\omega)) - \sigma(\pi_{k^c-1}^I(\omega))] \\
\leq &e^{\frac{(\epsilon-1) \cdot k^c}{k}} \cdot \mathbb{E}_\omega[\sigma(\pi'(\omega)) - \sigma(\pi_0^I(\omega))] \\
= &e^{\frac{(\epsilon-1) \cdot k^c}{k}} \cdot \mathbb{E}_\omega[\sigma(\pi'(\omega))].
\end{aligned}
$$

By rearranging it, we have

$$\mathbb{E}_\omega[\sigma(\pi_{k^c}^I(\omega))] \geq (1 - e^{\frac{(\epsilon-1) \cdot k^c}{k}}) \cdot \mathbb{E}_\omega[\sigma(\pi'(\omega))]. \tag{21}$$

In addition, $\mathbb{E}_\omega[\sigma(\pi'(\omega))] \geq \mathbb{E}_\omega[\sigma(\pi^*(\omega))]$ holds, as a certain number of budgets are necessary for meeting the threshold requirement. Combining this with Eq. (21), we have

$$\mathbb{E}_\omega[\sigma(\pi_{k^c}^I(\omega))] \geq (1 - e^{\frac{(\epsilon-1) \cdot k^c}{k}}) \cdot \mathbb{E}_\omega[\sigma(\pi^*(\omega))]. \tag{22}$$

Combining Eq. (22) with Eq. (20), the theorem follows. □

**PROOF OF THEOREM 4.3.** Recall that in each iteration, AAS selects one of the two greedy-based procedures to execute based on a specific judgment condition, i.e., $T - I_P(S) \leq k - |S| - 1$. To explain this inequality, $T - I_P(S)$ means the number of prioritized nodes that still need to be activated currently, and $k - |S|$ indicates the remaining budget currently. Besides, it is worth noting that executing the procedure in Lines 8-10 once guarantees a gain of at least 1 for $I_P(S)$.

When $T - I_P(S) \leq k - |S| - 1$ holds (Line 3), the procedure for maximizing the influence spread in $G$ is executed. Then, we will make an observation for the diffusion results. If there is no prioritized node activated (in the worst case), that is, $I_P(S)$ remains unchanged and $|S|$ increases by 1, $T - I_P(S) \leq k - |S|$ holds. In such a scenario, the threshold condition can still be satisfied by consistently executing the procedure in Lines 8-10, if necessary. In contrast, when $T - I_P(S) = k - |S|$ holds (Line 7), it is essential to implement the procedure for maximizing $I_P(\cdot)$, otherwise we may fail to achieve $I_P(S) \geq T$. By combining these two cases, the theorem follows. □

**PROOF OF THEOREM 4.4.** According to [14], the expected time complexity of the procedure in Lines 4-6 is $O((\log n + \log \frac{1}{\epsilon})(m + n)/\epsilon^2)$. In addition, since the expected time required to generate a PRR set is $\frac{\mathbb{E}[I_P(v^*)]}{|P|} m$, and the number of generated PRR sets is $\theta_p$, the expected time complexity of the procedure in Lines 8-10 is $\frac{\mathbb{E}[I_P(v^*)]}{|P|} m \theta_p$. Since there are $k$ iterations within AAS, the theorem holds. □

