# OpenReview forum: "Effective Influence Maximization with Priority"
_ACM.org/TheWebConf/2025/Conference — WWW 2025 Poster_

### Official Review · Reviewer_KpXK · 2024-12-01

**Novelty:** 2
**Technical Quality:** 3

**Review:**

This paper focuses on IMP, a variant of the classic IM problem in social networks. The authors propose a two-stage SAR framework and further expand it to the adaptive setting.

Strengths:

- The paper is written well and easy to follow.

- The authors present a feasible solution to the IMP problem.

- Theoretical analyses in the appendix.

Weaknesses:

- The IMP problem is not new and has been investigated by an existing work [24]. This paper shows an improved work to the existing solution, with less originality.

- The authors did not present a comprehensive review of the related work. The classification of non-adaptive IM and adaptive IM cannot showcase the latest representative works about IM and its variant problems.

- The experimental validation is less convincing when compared with only a single method.

**Questions:**

- The IMP problem does not seem to exist in real-world scenarios, and the example provided by the authors regarding the promotion of gaming equipment is unconvincing. First, what exactly does "activation" refer to in this example? If it refers to purchase behavior, why must the purchases by a group of users with special priority reach a certain threshold? Moreover, where do these special-priority users come from? If the company has a set of special-priority users, it can divide the promotion into special-priority user promotion and ordinary user promotion.

- According to the two-stage setting, many existing solutions to the IM problem can also be used to address the IMP problem.

- Besides IGS, have the authors considered adapting some classic and latest IM solutions to the IMP problem for comparison?

**Reviewer Confidence:**

4: The reviewer is certain that the evaluation is correct and very familiar with the relevant literature

**Scope:**

2: The connection to the Web is incidental, e.g., use of Web data or API

---

### Official Review · Reviewer_JjF7 · 2024-12-01

**Novelty:** 5
**Technical Quality:** 6

**Review:**

This paper studies the influence maximization with priority (IMP) problem, where the users in the propagation network are not equally important. The authors identified that existing IM solutions are suboptimal for IMP and can only have a poor approximation ratio. Specifically, (1) the two-stage solution puts too much budget into the first stage; (2) determining the appropriate/optimal budget for the first stage is challenging. Then the authors proposed a Select-And-Replace (SAR) framework, which is also a two-stage algorithm (select stage and replace stage). It is showed that SAR can offer a better approximation and a larger influence spread. Moreover, the authors extended the non-adaptive setting in IMP. To make the SAR framework suitable for the adaptive setting, the authors designed an Adaptive-Alternation-Selection (AAS) framework. The SAR and AAS frameworks were evaluated on five real-world datasets, the results showed that the performance of the proposed frameworks is better than the state-of-the-art baselines, in terms of influence spread and empirical approximation ratio.

Pros:
1. The IMP problem is an important new setting for the real-world applications of viral marketing.
1. The proposed frameworks achieve better approximation ratios and larger influence spreads.
1. The performance of the proposed frameworks surpasses that of the state-of-the-arts on five real-world datasets. The authors also provided theoretical guarantees.

Cons:
1. The relationships between the proposed frameworks and weighted/targeted IM solutions should be better explained/illustrated.
1. The limitations of the proposed frameworks should be discussed in the paper.

**Questions:**

Please see the cons.

**Reviewer Confidence:**

2: The reviewer is willing to defend the evaluation, but it is likely that the reviewer did not understand parts of the paper

**Scope:**

3: The work is somewhat relevant to the Web and to the track, and is of narrow interest to a sub-community

---

### Official Review · Reviewer_F3oN · 2024-12-02

**Novelty:** 5
**Technical Quality:** 5

**Review:**

This paper addresses the problem of influence maximization, a central problem in social network analysis. In particular, the paper studies the variant of the problem where we have priority constraints, a variant proposed recently in 2020. The authors identify limitations in existing solutions and propose two novel algorithms.

1. SAR (Select-And-Replace): A non-adaptive algorithm that first selects influential nodes to maximize overall spread, then strategically replaces nodes to meet priority thresholds while maintaining strong theoretical guarantees.

2. AAS (Adaptive-Alternation-Selection): An adaptive algorithm that dynamically alternates between maximizing overall influence and meeting priority requirements based on observed diffusion results.

The approximation ratio of SAR depends on the actual budget within the solution of SAR that maximizes the influence spread in G, thus strictly speaking does not provide a provable worst-case constant-factor approximation guarantee. However, on real-world datasets, SAR does show distinctly better performance compared to IGS, the previous state-of-the-art. AAS, on the other hand, is the first algorithmic result for the IMP problem under the adaptive setting, and is shown in the paper to perform even better empirically.

Overall, the paper makes interesting progress on a variant of the central influence maximization problem. The algorithms proposed in the paper do not have worst-case constant approximation ratio, but do perform better in practice. It would be nice to see either a true approximation algorithm for the problem, or a better hardness result.

**Questions:**

- Is the full name of IGS formally defined before the abbreviation appears for the first time in the introduction?

- Is there any evidence suggesting IMP is distinctly hard to approximate theoretically than IM?

**Reviewer Confidence:**

3: The reviewer is confident but not certain that the evaluation is correct

**Scope:**

4: The work is relevant to the Web and to the track, and is of broad interest to the community

---

### Official Review · Reviewer_9j3H · 2024-12-02

**Novelty:** 5
**Technical Quality:** 6

**Review:**

This paper presents a novel exploration of the Influence Maximization with Priority (IMP) problem, proposing the Select-And-Replace (SAR) framework for the non-adaptive setting and the Adaptive-Alternation-Selection (AAS) framework for the adaptive setting. The contributions are clearly stated, and the theoretical foundations are strong, with guarantees provided for the performance of both SAR and AAS. Comprehensive experiments on real-world datasets substantiate the effectiveness of the proposed methods, showing significant improvements over state-of-the-art techniques.

Strengths:

1. The paper demonstrates solid theoretical foundations. Both SAR and AAS provide provable approximation guarantees, which are rigorously analyzed.

2. Extensive experiments across multiple datasets validate the effectiveness of the proposed methods. Both SAR and AAS show superior performance in terms of influence spread and approximation ratio compared to existing methods.

3. Addressing IMP in the adaptive setting through the AAS framework is innovative and practical, reflecting real-world scenarios more accurately.

4. The paper is well-structured, with clear problem formulation, detailed algorithmic descriptions, and thoughtful experimental analysis.

Concerns:

Based on my understanding, the AAS framework, while effective, might face challenges in parameter selection for the adaptive setting in different contexts, which could have been discussed in more detail.

**Questions:**

See the concerns.

**Reviewer Confidence:**

3: The reviewer is confident but not certain that the evaluation is correct

**Scope:**

4: The work is relevant to the Web and to the track, and is of broad interest to the community

---

### Official Review · Reviewer_64aR · 2024-12-02

**Novelty:** 6
**Technical Quality:** 7

**Review:**

#### **Overall Assessment**
The paper introduces two novel algorithms for the problem of Influence Maximization with Priority (IMP), offering data-dependent approximation guarantees. These algorithms improve the state-of-the-art in the non-adaptive setting and provide tailored solutions for the adaptive setting. The proposed methods, theoretical analysis, and experimental validation align with the high standards of *The Web Conference* and make a contribution to the field of influence maximization. The following detailed review highlights the paper’s strengths and areas for improvement.

---

### **1. Quality**

- **Strengths**:
    - The proposed algorithms set a new state-of-the-art for the IMP problem, addressing both non-adaptive and adaptive settings.
    - The theoretical analysis is rigorous and robust, offering strong support for the proposed approaches.
    - Extensive experimental evaluations demonstrate the effectiveness of the methods across diverse datasets.
    - The paper is well-written and logically structured, making the content accessible to the reader.

- **Areas for Improvement**:
    - **Line 382 (left column)**: It is not straightforward to understand the intuition behind the necessity of another set \( R_2 \), given that \( R_2 \) is sampled in the same way as \( R_1 \). Please provide a clear explanation of its purpose and role in Eqs. (4) and (5).
    - **Line 446 (right column)**: Clarify the reasoning behind processing the seeds in reverse order. An intuitive explanation would greatly enhance understanding.
    - **Line 490 (left column)**: Is \( k_1^b \) returned by SAR consistently larger than \( k_b^a \) from IGS due to a theoretical reason or is this purely an empirical observation? Please specify.


---

### **2. Clarity**

The paper is generally clear and well-structured. The algorithms are well-introduced, and limitations of existing literature are effectively outlined. The following enhancements could improve clarity further:

- **Notation Table**: Consider integrating a table summarizing key notations and concepts to reduce the need for cross-references. This would be particularly useful for readers navigating the appendix.
- **Proofs in Theorem 3.1**: Clearly state the substitutions made in the proof (e.g., \( \lambda = ... \)) to streamline understanding, especially when transitioning from Eq. (10-11) to Eqs. (12) and (13).
- **Captions**: Make captions more informative and self-contained to enable readers to interpret figures and tables without extensive cross-referencing.

---

### **3. Originality**

The paper builds on existing influence maximization techniques but introduces two novel algorithms tailored to the IMP problem, including the adaptive setting. This represents a meaningful extension of the literature, with both theoretical and practical contributions.

---

### **4. Significance**

The paper is highly relevant to the field of influence maximization on social media and the track.

---

### **5. Reproducibility**

- While the repository is not provided, the pseudocode for the algorithms is detailed and self-explanatory, which aids reproducibility.
- Sharing the implementation and experimental scripts in a public repository would significantly enhance reproducibility and facilitate broader adoption of the proposed methods.

---

**Questions:**

Please address the points outlined in the review.

**Reviewer Confidence:**

2: The reviewer is willing to defend the evaluation, but it is likely that the reviewer did not understand parts of the paper

**Scope:**

4: The work is relevant to the Web and to the track, and is of broad interest to the community